# Rehabilitation interventions to modify physical frailty in adults before lung transplantation: a systematic review protocol

Laura McGarrigle [1,2] Gill Norman [2,3] Helen Hurst,[4,5] Chris Todd [2,6]

[1]Cardiothoracic Transplantation, Manchester University NHS Foundation Trust, Manchester, UK
[2]National Institute for Health and Care Research Applied Research Collaboration Greater Manchester, Greater Manchester, UK
[3]Division of Nursing, Midwifery & Social Work, University of Manchester, Manchester, UK
[4]School of Health and Society, University of Salford, Salford, UK
[5]Renal, Salford Royal Hospital, Northern Care Alliance NHS Foundation Trust, Salford, UK
[6]School of Health Sciences, University of Manchester, Manchester, UK

**Correspondence to**
Mrs Laura McGarrigle;
laura.mcgarrigle@mft.nhs.uk

## ABSTRACT

**Introduction** Lung transplantation is the gold-standard treatment for end-stage lung disease for a small group of patients meeting strict acceptance criteria after optimal medical management has failed. Physical frailty is prevalent in lung transplant candidates and has been linked to worse outcomes both on the waiting list and postoperatively. Exercise has been proven to be beneficial in optimising exercise capacity and quality of life in lung transplant candidates, but its impact on physical frailty is unknown. This review aims to assess the effectiveness of exercise interventions in modifying physical frailty for adults awaiting lung transplantation.

**Methods and analysis** This protocol was prospectively registered on the PROSPERO database. We will search four databases plus trial registries to identify primary studies of adult candidates for lung transplantation undertaking exercise interventions and assessing outcomes pertaining to physical frailty. Studies must include at least 10 participants. Article screening will be performed by two researchers independently at each stage. Extraction will be performed by one reviewer and checked by a second. The risk of bias in studies will be assessed by two independent reviewers using tools appropriate for the research design of each study; where appropriate, we will use Cochrane Risk of Bias 2 or ROBINS-I. At each stage of the review process, discrepancies will be resolved through a consensus or consultation with a third reviewer. Meta-analyses of frailty outcomes will be performed if possible and appropriate as will prespecified subgroup and sensitivity analyses. Where we are unable to perform meta-analysis, we will conduct narrative synthesis following Synthesis without Meta-analysis guidance. The review will be reported using the Preferred Reporting Items for Systematic Reviews and Meta-Analyses checklist.

**Ethics and dissemination** No ethical issues are predicted due to the nature of this study. Dissemination will occur via conference abstracts, professional networks, peer-reviewed journals and patient support groups.

**PROSPERO registration number** CRD42022363730.

## STRENGTHS AND LIMITATIONS OF THIS STUDY

⇒ Rigorous systematic review methods at all stages of the review combined with clinical expertise will allow us to produce a reliable first synthesis of the evidence for the effectiveness of rehabilitation in lung transplant candidates for physical frailty.

⇒ A comprehensive search for relevant studies, with input from an information specialist, from multiple databases and other sources will allow us to identify relevant studies wherever published.

⇒ The exclusion of non-English-language studies is a limitation of this study but we will list these studies where we identify them.

⇒ It is possible that we may miss some studies of individuals with chronic lung disease with relevant data, where the inclusion of lung transplant candidates is not specified by the authors, but this is unlikely to substantively impact the review outcomes.

⇒ Using outcomes as a key criterion for inclusion risks missing some relevant studies due to the potential for reporting bias; to mitigate this, we will attempt to contact authors of all otherwise relevant studies to establish if any further outcomes were assessed but not reported and, where possible, obtain relevant data.

a range of lung diseases including chronic obstructive pulmonary disease (COPD), interstitial lung diseases, pulmonary arterial hypertension and cystic fibrosis (CF). Poor lung function is associated with reduced exercise tolerance, dyspnoea and disability. The terminal stages of these conditions have a profound impact on an individual's physical function and quality of life. LTx is a well-established therapy for chronic lung disease in a very specific population who meet the stringent, internationally accepted criteria.[1] The strict criteria are necessary to optimise outcomes and protect from inappropriate allocation of such a scarce resource. On average, there are around 350 adults on the active LTx waiting list in the UK. In the 10 years prior to the COVID-19 pandemic, the

## INTRODUCTION
### Lung transplantation

Lung transplantation (LTx) involves surgically replacing diseased organs in individuals with advanced respiratory failure due to

mean number of lung transplants performed annually in the UK was 178 (±18).[2]

## Frailty

Frailty is a state characterised by lack of physiological reserve and increased vulnerability to stressors and is common in chronic end-stage lung disease,[3 4] particularly in those referred for LTx.[5] There has been a significant rise in the proportion of LTx recipients aged over 65 years,[6] and increasing age is an independent risk factor for both poor outcomes after LTx[7] and increased incidence of frailty.[8]

The two main models conceptualising frailty are the phenotypic and cumulative deficit models. The phenotypic model is more commonly used in LTx and considers frailty through five characteristics: shrinking (weight loss), weakness, exhaustion, slowness and low physical activity.[9] The cumulative deficit model conceptualises frailty as an accumulation of symptoms, comorbidities, diseases and health deficiencies. The greater the number of 'deficits', the higher the frailty of the individual.[10] Although previously linked to decreased survival after LTx,[11] the validity of the latter model in predicting other LTx outcomes is unclear.[5]

Although psychosocial, nutritional and physical impairments impact post-LTx outcomes,[1] the clinical operationalisation of frailty measurement in LTx relates overwhelmingly to the physical domain.[5 12 13] This may be due to the feasibility of data collection, the younger age of LTx candidates or the complexity associated with cumulative deficit models.[12 14] The most commonly reported phenotypic frailty measures collected in clinical practice prior to LTx include the Short Physical Performance Battery (SPPB) and the Fried Frailty Phenotype (FFP).[13 15]

Research studies in LTx reflect this phenotype approach. Although conceptualisation and investigation of biomarkers and their links to frailty[16] (including in patients with lung disease) have recently been undertaken, the field has lacked established markers or cut-offs, preventing their routine use in the evaluation of frailty for LTx.[5 12 13] The recent development of an LTx-specific frailty scale, the Lung Transplant Frailty Scale (LT-FS) and its validation in 342 LTx candidates may help to resolve this.[13] This uses clinically feasible measures of body composition (including appendicular skeletal muscle index and per cent body fat by bioelectrical impedance analysis), research-grade serum biomarkers and clinical laboratory blood markers in addition to measurement of physical function (SPPB and FFP). All models of the scale displayed superior predictive validity for short-term mortality and removal from the LTx waiting list than the FFP or SPPB alone, but long-term outcomes have not yet been evaluated.

Previous reviews have identified physical frailty as being detrimental to morbidity and mortality both before and after LTx.[17] Physical frailty is also associated with an increased risk of readmission after LTx.[18] Transplant teams experience the challenges of identifying patients with the physical and psychological reserve necessary to survive such a demanding perioperative and postoperative period and to thrive long term.[1 12] This challenge has been exacerbated by the lack of consensus on the optimal assessment tools for frailty in this field,[1] which the LT-FS may help to resolve.[13] Despite the current lack of consensus on tool utility, it is widely agreed that further work is required to establish the effectiveness of interventions to modify physical frailty, improve candidate selection and LTx outcomes.[1 11 13]

## Exercise and rehabilitation prior to LTx

Rehabilitation for individuals with lung conditions usually takes the form of pulmonary rehabilitation, an evidence-based programme of exercise interventions and education. It aims to reduce dyspnoea, optimise functional capacity, increase participation, and reduce healthcare costs through exacerbation and hospital admissions.[19] Rehabilitation while on the waiting list (sometimes known as prehabilitation) is recommended for all LTx candidates.[1] Rehabilitation after LTx is also a vital component of the recovery programme after surgery and intensive care unit (ICU)/hospital stay.[20] The course of the SARS COVID-19 pandemic has seen many pulmonary rehabilitation schemes develop virtual or telephone offers alongside a face-to-face programme. Technological advances have also led to the development of digital alternatives.[21] A recent European Society for Organ Transplantation consensus statement on prehabilitation for solid organ transplant candidates concluded that 'it is feasible, acceptable and safe for adults to participate in exercise, nutritional and pscychosocial interventions during the waiting list period' (Annema et al,[22] p.19). The importance of multimodal programmes was highlighted but the lack of a core outcome set and specific guidance on programme content highlights the need for ongoing, high-quality research studies in this field.

## Potential impact of the intervention

Advancing lung disease impacts on exercise capacity and accelerates muscle atrophy. Reduced muscle mass and quadriceps strength are observed in the pre-transplant and post-transplant period, with the impact of postoperative immunosuppressive therapy an important factor in the development of obesity, sarcopenia and osteoporosis.[23 24] A recent systematic review concluded that exercise programmes containing aerobic and resistance training prior to LTx have the capacity to improve exercise capacity and quality of life with some evidence of increases in muscle strength.[20] Multicomponent exercise in older adults has been shown to contribute to an improvement in muscle strength, balance, lung function, physiological processes (such as reduced oxidative damage, inflammation and improved mitochondrial function) and physical activity which in turn impacts on muscle composition and function, strength, function in daily life and therefore physical frailty.[25] Pulmonary rehabilitation has been

shown to improve FFP scores towards a more robust state in the short term in individuals with COPD.[26]

## Why it is important to do this review

Professionals in the transplant community are increasingly measuring frailty in attempts to evaluate surgical suitability and weigh the risks and potential benefits of LTx for older individuals. There is emerging evidence linking frailty to reduced quality of life and mortality on the lung transplant waiting list and postoperative adverse outcomes including increased length of hospital stay, readmission, disability and worse health-related quality of life.[5] Frailty is therefore a significant concern for transplant programmes.

The aim of LTx is to improve survival, function and quality of life of the recipient, while considering the ethical elements of who will benefit from such a limited pool of donor organs.[12] Understanding and targeting reversible components of frailty has the potential to improve physical condition ahead of major surgery, reduce waiting list mortality and potentially impact postoperative outcomes and maximise benefit from LTx.[4 12 27] Identifying the best strategies to improve frailty prior to transplantation has been highlighted as an area that requires investigation.[4 5 13]

Previous systematic reviews have documented the beneficial effect of rehabilitation on exercise capacity and quality of life prior to a lung transplant.[20 28] However, there are no prior reviews specifically addressing interventions in modifying physical frailty in this population despite the significance of the problem and link to poor outcomes.[17]

## Aims and objectives

This systematic review aims to evaluate the impact of exercise interventions in modifying physical frailty for adults awaiting LTx. We also aim to identify any harms that occur as a result of an exercise intervention.

## METHODS AND ANALYSIS
### Study design

This study will be a systematic review with methodology following guidance from the Cochrane Handbook of Systematic Reviews.[29] It will be reported in accordance with the Preferred Reporting Items for Systematic Reviews and Meta-Analyses checklist.[30] Any important amendments will be documented on PROSPERO.

### Inclusion and exclusion criteria
#### Types of studies

Preliminary searches indicated very few randomised controlled trials (RCTs) and non-RCTs, which we initially planned to include. We therefore widened the inclusion criteria to original primary research studies with any design, including those without controls, with more than 10 participants; this amendment was registered on PROSPERO. We therefore anticipate including randomised and non-randomised trials, cohort studies and case series. Cross-sectional studies may be eligible if they assess frailty following an intervention. We will not include evidence syntheses, commentaries, case studies and non-systematic narrative reviews. Where possible, we will extract and report any definitive statements regarding ethical procurement of donor organs and exclude reports from institutions with unethical organ harvesting practices.

### Types of participants

We will include studies of adults over the age of 18 years, on a waiting list (candidates) for a single or double LTx. Individuals with any underlying lung disease who have met the criteria to be accepted onto the waiting list at an LTx centre, where frailty has been assessed, will be included. There are well-established and accepted internationally agreed criteria for LTx.[1] We will include studies involving candidates for primary or subsequent repeat LTx procedures. LTx performed by any surgical incision site will be included. Studies of candidates listed for a multiorgan transplant in the same surgical procedure (including but not limited to, concurrent heart and lung or lung and liver) will be excluded unless data are provided for lung recipients only or multiorgan candidates comprise under 25% of study. We will exclude studies of people completing interventions while receiving extra corporeal membrane oxygenation as a bridge to transplant; they are restricted to interventions within the ICU which is not the focus of this review.

### Types of intervention

We will include any formal physical exercise or physical activity prescribed under professional guidance which includes formal evaluation of outcomes. The intervention may be supervised or unsupervised, face to face or virtual and performed in any setting (community or hospital). There is no minimum length or intensity of intervention. The intervention may be single (one form of exercise), multimodal (for example, a combination of strength and endurance training) or multicomponent (eg, exercise and a nutritional intervention). The intervention must contain an exercise or physical activity component. Physical activity is defined as any bodily movement produced by skeletal muscles that results in energy expenditure. Exercise is defined as a subset of physical activity that is planned, structured and repetitive and has an objective of the improvement or maintenance of physical fitness.[31] Multimodal and multicomponent interventions in this population are clinically important due to the need to address the complex interaction between the physical and psychological health of the patient prior to transplantation.[22]

### Types of comparators

We will accept any comparator or no comparator. Where studies include comparison groups, we anticipate identifying comparisons of rehabilitation with one or more of the following: no intervention, 'usual care', advice only,

or an alternative intervention which may or may not meet our intervention criteria. The specific rehabilitation intervention should be the only systematic difference between the groups.

## Outcomes

We will consider any validated physical frailty measures as primary outcomes: measures of phenotypic frailty (FFP) or cumulative deficit frailty models (eg, Clinical Frailty Scale, Electronic Frailty Index).

We will also include surrogate markers of physical frailty, for example, SPPB, sarcopenia via hand grip dynamometry, quadriceps force, CT scan (muscle cross-sectional area), muscle strength testing of upper of lower limb: manual or non-manual (eg, 1 rep max), sit to stand testing and objective assessments of balance.

Studies must include at least one direct or indirect measure of frailty, measured before (except in the case of cross-sectional studies) and at any time point after the intervention in order to be eligible for inclusion. We will specify whether outcomes have been assessed preoperatively or postoperatively and discuss any post-LTx rehabilitation (as a potential confounder) where relevant. Where no relevant outcome is reported, we will attempt to contact authors of all otherwise relevant studies to establish if any further outcomes were assessed but not reported and, where possible, obtain relevant data. We will distinguish between studies excluded after confirmation that no relevant outcomes were assessed and those where this confirmation was not possible.

Where studies report a relevant primary outcome, we will consider the following as secondary outcomes: mortality (on waiting list or postoperatively), hospital/ICU length of stay, health-related quality of life measures. Any adverse events reported during interventions will be recorded.

## Search strategies

Literature search strategies have been developed using medical subject headings related to rehabilitation, exercise and lung transplant candidates with support from an experienced medical librarian. Trials will be identified from searches of the following databases: MEDLINE (Ovid) 1980 to date, EMBASE (Ovid) 1980 to date, CINAHL Plus (EBSCO) 1980 to date, Cochrane Central Register of Controlled Trials, the Cochrane Library, trial registries (ClinicalTrials.gov and the WHO trials portal). All databases will be searched from 1980 to the present. The success of LTx was established only after the discovery and introduction of the immunosuppressive agent ciclosporin which became accepted practice in the early 1980s.[32] A draft search strategy for MEDLINE is given in table 1 with this strategy being adopted and modified for other databases (see online supplemental material). We will check if abstracts meeting the inclusion criteria have since been published in full text. We will exclude theses or unpublished studies. In an attempt to reduce our susceptibility to reporting bias, we will provisionally include all studies

**Table 1** Ovid MEDLINE(R) search strategy

| Number of search | Search term |
| --- | --- |
| 1 | lung transplant*.ti,ab. |
| 2 | exp Lung Transplantation/ |
| 3 | 1 OR 2 |
| 4 | (wait* or candidate* or pre op* or pre-op* or await*).ti,ab. |
| 5 | Waiting Lists/ |
| 6 | Preoperative Exercise/ or Preoperative Care/ |
| 7 | 4 or 5 or 6 |
| 8 | (exercis* or rehab* or prehab*).ti,ab. |
| 9 | exp Rehabilitation/ |
| 10 | 8 OR 9 |
| 11 | 3 AND 7 AND 10 |
| 12 | limit 11 to (yr="1980 -Current" and "all adult (19 plus years)") |

meeting our inclusion criteria for population, intervention, comparator and study design. If frailty outcomes are not reported, we will contact authors to check if any were measured but excluded from the publication. We will not exclude studies based on outcomes in the first instance.

We will also search the references of all identified included studies and of identified systematic or scoping reviews.

## Selecting studies

Following deduplication in EndNote, title and abstract and full-text screening will be performed by two independent reviewers (LM and GN) against the predefined inclusion criteria using Rayyan software.[33] Any discrepancies between reviewers will be resolved by a consensus or consultation with a third reviewer (HH or CT). Any non-English-language studies will be retained and listed for reference but not included in the synthesis process. This will enhance transparency, mitigate language publication bias and map the extent of missing evidence in our results.

## Data extraction and management

Two authors will pilot and agree a data extraction form. Data extraction will be completed by one author (LM) and checked by a second (GN). Any discrepancy will be reconciled by a consensus or a third author (HH or CT). The data extraction form will be based on the information collated in table 2.

Where outcomes use continuous scales of measurement (eg, composite frailty scores, muscle cross-sectional area), we will express the results as the mean difference or, where scales or units are unclear or composites, standardised mean difference (SMD) with 95% CIs. Where outcome measures vary across studies, we will express them as SMD if we consider that they are assessing the same outcome construct. We will express dichotomous

**Table 2** Data extraction plan

| | Data to be extracted |
|---|---|
| Study design | Design (eg, RCT)<br>Date of study period and publication<br>Country of origin and setting, for example, single-site or multicentre study<br>Duration of follow-up<br>Sample size calculation<br>Ethical approval and (methods of) informed consent, IRB number<br>Any statement of ethical procurement of donor organs or reports of donor consent status |
| Participants | Inclusion/exclusion criteria<br>Characteristics for each group including:<br>Number of participants<br>Age (mean (SD))<br>Ratio of male:female<br>Disease/diagnosis<br>Comorbidities<br>Primary transplant (or repeat transplant after graft failure); number/percentage per group<br>Characteristics that stratify health opportunities and outcomes (PROGRESS-plus)[41] |
| Interventions | Based on Template for Intervention Description and Replication Checklist[42] and the Consensus on Exercise Reporting Template[43]<br>Type of exercise component or modality/other interventions in each group<br>Specific exercises and target muscle groups trained<br>Equipment required or provided<br>Frequency, intensity and duration of intervention (including how this was prescribed and measured)<br>Location of intervention delivery (eg, home, hospital, community, gym)<br>Who delivered the intervention (profession, level of experience, training)<br>Mode of intervention delivery (eg, face to face, telephone, video call)<br>Any reported tailoring, modification or personalisation of the intervention<br>Adherence measurement methods and outcomes (including adherence to attendance and exercise prescription)<br>Details of any motivation or behavioural change strategies used |
| Outcomes and measures of effect | Outcomes: any validated measure or surrogate measure of frailty including, but not limited to: FFP, Clinical Frailty Scale, Electronic Frailty Index, SPPB, hand grip strength, measures of muscle strength/size/force, sit to stand testing, measures of balance<br>Definition for each outcome alongside minimal clinically important difference where available, scale limits and direction of benefit<br>Time points measured<br>Validation of tool. For each group and each outcome at each time point, we will extract the number of participants with data in each group. We will extract post-intervention and change from baseline data.<br>Continuous outcomes: mean with SD for each group. Medians with IQR will be extracted where reported. Mean differences or standardised mean differences with 95% CI will be extracted where these are the only reported data. P values will be extracted in the absence of other outcome data as will descriptive reporting of results.<br>Dichotomous outcomes: number in each group with event. For adverse events, types of adverse events will also be reported. Relative risks or ORs with 95% CI will be extracted where these are the only reported data.<br>Time to event data: HRs or data to calculate these |
| Other | Other relevant details of study design or intervention<br>Funding source<br>Conflicts of interest<br>Recruitment failure<br>Patient and public involvement or engagement |

FFP, Fried Frailty Phenotype; IRB, Institutional Review Board; RCT, randomised controlled trial; SPPB, Short Physical Performance Battery.

outcomes as risk ratios with 95% CI. Survival outcomes will be reported as HRs with 95% CI. If outcomes are reported at multiple time points, we will extract all the data but will analyse primary data at the end of the intervention and at the latest time point reported. We anticipate that a mix of final and change from baseline scores will be available and will extract both or either, as mean differences with SD (see data synthesis).

## Missing data

Any missing data will be requested from the original author by email and included in the review. If we are

unable to obtain the missing data, where possible, we will calculate or impute these (eg, where measures of variance are not available). Where outcome data remain missing and cannot be imputed, we will use assumptions in analysing dichotomous data; that is, we will assume that missing participants did not have the outcome assessed, but we will explore the impact of this assumption with a sensitivity analysis. For continuous outcomes, we will perform a complete case analysis and will not attempt to impute results for missing participants.

## Risk of bias assessment

RCTs and quasi-RCTs will be assessed using the Cochrane Risk of Bias tool with an extension for cluster or crossover RCTs as required.[29] ROBINS-I[34] or another appropriate tool will be used for non-randomised controlled studies. Other study designs (eg, cohort and cross-sectional studies and case series) will be assessed using an appropriate tool. Risk of bias assessments will be done by two reviewers independently (LM and GN) and any discrepancies reconciled by a third reviewer (HH or CT).

## Data analysis: strategy for data synthesis

We will take a pragmatic approach to data management and synthesis based on the data available and the designs of included studies. Clinical and methodological heterogeneity will be considered initially. If sufficient studies are identified with low heterogeneity, statistical heterogeneity will be considered to determine if a random-effects meta-analysis should be performed. We will not combine randomised and non-randomised studies in the same analysis but may display them using a single forest plot. We anticipate that we will identify studies reporting one or both of post-intervention data and change from baseline data. Decisions on which of these we use in analyses will be guided by pragmatic considerations; while we may combine the two types of data if mean differences are calculated, we will not do so if SMDs are used. We will fully document these choices. If change from baseline data is presented without final score data, we will use an approach suggested in the Cochrane Handbook[28] to estimate the SD of the final score using an estimated correlation coefficient.

If the studies identified vary widely in terms of population, study design, intervention and outcomes, a meta-analysis may be inappropriate and therefore a narrative synthesis of findings will be produced and reported following the Synthesis without Meta-analysis guidelines for narrative evidence synthesis.[35] If we are unable to perform meta-analysis, we will present studies where we can compute a comparable effect size on a forest plot without combination. Where it is not possible to do this, we will report the variety of reported effect sizes in a table with CIs and may also display effect directions.

## Assessment of heterogeneity

We will initially assess statistical heterogeneity by visual inspection of the forest plot followed by a quantification using the $I^2$ statistic (75–100% considerable heterogeneity; 50–90% considered to represent substantial heterogeneity; 30–60% moderate and 0–40% considered unimportant heterogeneity).[29]

## Subgroup analysis

Participant heterogeneity may be investigated using subgroup analysis by disease type (COPD, CF, Idiopathic Pulmonary Fibrosis (IPF), for example). Subgroup analysis by age may be considered (18–69 and ≥70 years) as lung transplant recipients over the age of 70 years have a decreased longer-term survival.[36] Intervention heterogeneity may be investigated by subgroup analysis including the following potential groups: virtual and in-person rehabilitation, exercise intensity, exercise type (eg, aerobic vs resistance) or the degree of supervision (eg, fully supervised vs unsupervised). We will exercise caution in implementing these prespecified analyses and are mindful of the risks of performing multiple such analyses.

## Sensitivity analysis

If appropriate, we will use sensitivity analyses to explore the impact of any imputation of or assumptions about missing data. If we perform a meta-analysis with very few studies, we will conduct a fixed-effects analysis to assess the impact of the random-effects model in this context.[37]

## Summary of findings tables

We will present a summary of the main findings of the review in a table format including key information regarding the magnitude of effects for each outcome alongside a rating of the certainty of evidence. We will use the Grading of Recommendations Assessment, Development and Evaluation (GRADE) working group methodology considerations (risk of bias, consistency of effect, imprecision, indirectness and publication bias) to assess the quality of the body of evidence for each prespecified outcome.[38 39] We will justify all decisions to downgrade or upgrade the certainty of the evidence. If appropriate, we will present the evidence in summary of findings tables generated using GRADEpro software.[40]

## Patient and public involvement

Local lung transplant support group attendees have been involved in some preliminary patient and public engagement work prior to the data collection for this study. Further patient and public involvement work will be completed to contribute to the production of a plain language summary and insight into dissemination strategies.

## ETHICS AND DISSEMINATION

Ethical approval is not required because this is a systematic review using previously published data. We are mindful of the ethical considerations relating to the generation of these data and will record relevant information relating to this.

This systematic review has been registered prospectively (registration number CRD42022363730). Its findings will be submitted to international transplantation conferences and peer-reviewed journals for dissemination of the results and conclusions. Results and their implications will be shared at local and national meetings of LTx clinicians. We will also share findings with people living with chronic lung disease on the transplant waiting list and their carers via local patient support groups.

**Acknowledgements** The authors thank Bethan Morgan (librarian at Manchester University NHS Foundation Trust) for her support with electronic search strategies for this review and Sarah Rhodes (medical statistician) for her support with the meta-analysis approach.

**Contributors** LM had the idea for the review, wrote the first draft of the protocol and designed the search strategy. GN, HH and CT advised on the protocol, edited and commented substantively on the protocol. All authors approved the manuscript prior to submission.

**Funding** LM holds a part-time predoctoral fellowship funded by the National Institute for Health and Care Research Applied Research Collaboration Greater Manchester (NIHR ARC-GM) (grant award number NIHR200174). CT is a CI and partially funded and GN is wholly funded by NIHR ARC-GM.

**Disclaimer** The views expressed in this publication are those of the authors and not necessarily those of the NIHR or the Department of Health and Social Care or its partner organisations.

**Competing interests** None declared.

**Patient and public involvement** Patients and/or the public were involved in the design, or conduct, or reporting, or dissemination plans of this research. Refer to the Methods section for further details.

**Patient consent for publication** Not applicable.

**Provenance and peer review** Not commissioned; externally peer reviewed.

**ORCID iDs**
Laura McGarrigle http://orcid.org/0000-0002-2879-1970
Gill Norman http://orcid.org/0000-0002-3972-5733
Chris Todd http://orcid.org/0000-0001-6645-4505

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
