## [Reviewer comments · BMJ Open]

ARTICLE DETAILS

TITLE (PROVISIONAL)	Rehabilitation interventions to modify physical frailty in adults before lung transplantation: A systematic review protocol.
AUTHORS	McGarrigle, Laura; Norman, Gill; Hurst, Helen; Todd, Chris

VERSION 1 – REVIEW

REVIEWER	Fiatarone Singh, Maria University of Sydney, Faculty of Health Sciences
REVIEW RETURNED	16-Aug-2023

GENERAL COMMENTS	General Very well written but there are some issues with the search sensitivity and methodology and definition of data to be extracted. Specific comments: p 6 Line 26. "skeletal muscle mass" is not measured directly in body composition- do you mean LBM or FFM? p7. There are some issues with these statements: "Rehabilitation prior to LTx has been shown to improve exercise capacity and quality of life with some evidence of increases in muscle strength (21). Exercise has the potential to contribute to an increase in muscle strength, balance and physical activity which in turn impacts on physical frailty (23). Pulmonary rehabilitation has been shown to reverse frailty in the short term in individuals with COPD when assessed using the FFP (24)." Specifically: Muscle strength is rarely improved in programs that do not emphasize robust resistance training, which is generally the case. Frailty guidelines for older adults (eg Asia Pacific Guidelines) indicate that PRT should be the core of frailty -directed exercise. This needs to be discussed in the contxt of LTx, where sarcopenia/cachexia is prominent as a feature of frailty. Secondly not all exercise has the capability of improving strength and balance- these are modality-specific outcomes and aerobic exercise, which is common in prehab and rehap in fact rarely includes PRT. This needs to be evaluated and discussed specifically when reviewing papers for efficacy for frailty. Note that none of the FFP components is dependent on either aerobic capacity or aerobic exercise- with the exception of the level of physical activity. The other features are related to nutritoinal intake/status and strength/balance. "Reverse" frailty seems to strong a word. Is this actually the case?
---

	p.8. There is no mention of the fact that use of anti-rejection drugs post transplant leads to onset or worsening of sarcopenia, visceral obesity, osteopenia and metabolic syndrome/diabetes. This should be mentioned as another important reason for prehab exercise including PRT continued into rehab. Braith has shown since the 1980s that PRT prevents post-transplant related muscle wasting and osteoporosis. p 9 Please justify the age limit of 18+ p.9. Type of participants; given the well-known issues with organ harvesting under unethical circumstances in certain countries (using death row prisoners, political and religious prisoners, etc) and the prohibition of such activities or the publication of their results by most reputable transplantation journals, it is critical that a statement be included that no research data will be extracted from articles where it is not transparently clear that only ethical organ donation was the source of the lung transplants reported. Many guidelines now exist regarding this point, and it needs to be explicitly stated in all articles, including reviews, of transplant outcomes that such criteria were followed. p.10. More detail is needed on the exercise variables that will be collected- equipment, specific exercises used, muscle groups trained, modality, how intensity was measured, how adherence was measured, etc. p.11. It is not clear if you will include uncontrolled trials of multi-component interventions including exercise- for example exercise and nutrition. In such cases the independent effect of exercise cannot be determined for the outcome of frailty, which may improve due to many kinds of interventions. p.11. The only surrogate markers of frailty mentioned are physical ones What about weight loss and fatigue- the other 2 components of the FFP for example? Need to specify that any validated frailty tool will be included as there are dozens in use. p.12. The database search is quite limited. Should also include pre-medline, Sports Discus, AMED, PEDro, Web of Science and Google Scholar for better sensitivity p.13. The search is not sensitive enough. Need to search in all fields, not just title and abstract. Other terms for exercise should be included. In the limitations you say you searched for the outcome -this does not appear to be the case in the Medline example you included? Need to state explicitly that theses and unpublished studies will not be included. Need to state that abstracts will be followed up to see if published, as well as theses. p.14 Ethics approval needs to include the transparent reporting of the ethical nature of the organ donation process followed as this is routinely violated in publications from China. p.14 Exercise description needs to include type of equipment and specific exercises utilised, and how the intensity was measured and how the training was progressed if it was as well as any behavioural change methodology used. See CERT tool for the exercise elements that need to be included which are more specific than TIDieR for exercise studies. Adherence needs to
--	---

	include not only attendance, but adherence to the intensity, volume and progression principles of the prescription. p.15 by SMD, need to define the SD used- do you mean the Standardised mean response where it is the SD of the change score? Or Cohens d where it is the SD of the baseline sample? p.15. You have not defined the outcomes that will be included as index of frailty in the table, only how you will present them statistically. p. 17. You have not indicated how you will calculate ESs and how you will combine them if possible. What SD will you use? What will you do if only the SD of the change scores are reported, or only the post test scores are reported, and other studies report pre and post means and SDs? You will need relative ESs for the controlled trials and absolute ESs for the uncontrolled trials. Will you include studies that only measure frailty at post transplant assessment? You have not specified when the frailty outcomes must be measured relative to the intervention. Is it only if they are immediately pre and post intervention? What about followup post surgery? What about rehabilitation post- surgery? How will that be assessed and factored into the results? How soon after surgery can the frailty outcome be included as it is likely to change markedly in the peri-operative, acute hospital, post discharge phases. None of this is described and is critical to your findings.
--	--

REVIEWER	Janaudis-Ferreira, Tania McGill University, School of Physical and Occupational Therapy
REVIEW RETURNED	29-Aug-2023

GENERAL COMMENTS	Thank you for the opportunity to review this protocol for a systematic review and meta-analysis entitled "Rehabilitation interventions to modify physical frailty in adults before lung transplantation: A systematic review protocol". The protocol is well written and clear. The topic is of interest for the lung transplant community. My main concern is about the feasibility of this review. A systematic review on exercise interventions in solid organ transplant candidates, published by Pesce de Souza et colleagues in 2020, identified only one article in lung transplant candidates (a pre-post study) that included a measure of frailty. Although the authors address the issue of limited RCTs in this field in their protocol, and mention that they will consider study designs other than RCTs, I am still concerned that the number of published articles won't be enough for their review and meta-analysis. I suggest that the authors conduct a preliminary search including all study designs and present the results in their protocol. Another question I have is about the intervention of interest. Will the authors consider studies that included a multimodal pre-rehabilitation program? For example, that include a nutritional intervention alone or in addition to exercise? What about psychological interventions? A recent consensus statement from the European Society of Organ Transplantation (ESOT) (published in July 2023) focused on multimodal pre-habilitation interventions in SOT and may be a good reference for the authors. The reference list of this consensus statement may help the authors determine the feasibility of their review. https://pubmed.ncbi.nlm.nih.gov/37547750/
---

	“European Society of Organ Transplantation (ESOT) Consensus Statement on Prehabilitation for Solid Organ Transplantation Candidates” by Annema and colleagues. What is the minimum number of studies that the authors are considering for sub-group analysis and sensitivity analysis? Based on a preliminary search, the author may be able to know of these types of analysis will be feasible. Minor comment: Line 13, page 9 – a period is missing after the word “study”. This is definitely a very important topic and a methodologically sound protocol. I would encourage the authors to clarify these questions about feasibility and inclusion criteria before starting their review.
--	---

VERSION 1 – AUTHOR RESPONSE

Reviewer: 1 Maria Fiatarone Singh, University of Sydney		
p 6 Line 26. "skeletal muscle mass" is not measured directly in body composition- do you mean LBM or FFM?	The study in question (Singer et al, 2023) documented their assessment of body composition to include “skeletal muscle mass – appendicular skeletal muscle index (ASMI) – and percent body fat by bioelectrical impedance analysis”. The authors also measured and reported BMI and grip strength under the umbrella term of “body composition measures”. They justified the inclusion of these measures based on clinical feasibility. We have made changes to our manuscript to more accurately represent the exact measures reported by Singer et al.	Lines 87-90
p7. There are some issues with these statements: "Rehabilitation prior to LTx has been shown to improve exercise capacity and quality of life with some evidence of increases in muscle strength (21). Exercise has the potential to contribute to an increase in muscle strength, balance and physical activity which in turn impacts on physical frailty (23). Pulmonary rehabilitation has been shown to reverse frailty in the short term in individuals with COPD when assessed using the FFP (24)." Specifically: Muscle strength is rarely improved in programs that do not emphasize robust resistance training, which is generally the case. Frailty guidelines for older adults (eg Asia Pacific Guidelines) indicate that PRT should be the core of frailty -directed exercise. This needs to be discussed in the contxt of	We accept that generalised statements regarding “exercise” and frailty outcomes may be misleading and are cognizant that a need for brevity (due to word count limitations) may have impacted on the accuracy of our background and introduction. We have added some extra detail, particularly drawing on the paper by Angulo et al (2020) regarding the reviews/studies backing these statements. We plan to extract detailed information regarding the exercise components reported and delivered and will discuss these in relation to their efficacy of improving physical frailty if demonstrated by the included studies. We have increased the detail in Table 2 to reflect this. We thank the reviewer for these helpful comments as they highlight the importance of specificity as a training principle in regards to	From line 121 Table 2

LTx, where sarcopenia/cachexia is prominent as a feature of frailty. Secondly not all exercise has the capability of improving strength and balance- these are modality-specific outcomes and aerobic exercise, which is common in prehab and rehab in fact rarely includes PRT. This needs to be evaluated and discussed specifically when reviewing papers for efficacy for frailty. Note that none of the FFP components is dependent on either aerobic capacity or aerobic exercise- with the exception of the level of physical activity. The other features are related to nutritional intake/status and strength/balance.	modifying particular elements of frailty (e.g., balance, strength). We must ensure we discuss in detail, where possible, the training programmes provided and must highlight any lack of detail in intervention reporting in our findings.	
"Reverse" frailty seems to strong a word. Is this actually the case?	In the cited study by Maddocks et al. (2016), their population of pulmonary rehabilitation attenders were assessed according to the FFP criteria. Among the 115 completers who were frail prior to pulmonary rehabilitation, 71 (61.7%) were prefrail (64, 55.6%) or robust (7, 6.1%) following it. This demonstrates an improved frailty state according to the FFP definition. However, a small minority of completers, 13/390 (3.3%), had moved from a prefrail to a frail state after pulmonary rehabilitation. Despite the promising results in this study of frail individuals moving away from the FFP frail criteria, we agree that the term "reverse" could mislead the reader to assume all elements of the frailty syndrome had resolved. We have therefore adjusted the language within our manuscript to reflect this.	Lines 131-133
p.8. There is no mention of the fact that use of anti-rejection drugs post transplant leads to onset or worsening of sarcopenia, visceral obesity, osteopenia and metabolic syndrome/diabetes. This should be mentioned as another important reason for prehab exercise including PRT continued into rehab. Braith has shown since the 1980s that PRT prevents post-transplant related muscle wasting and osteoporosis.	We have added this important consideration to the manuscript.	Lines 122-124
p 9 Please justify the age limit of 18+	The concept of frailty and measurement in children has been explored in cardiac failure populations but there is limited evidence of the operationalisation of frailty measurement in paediatric chronic lung disease populations. Our scoping preliminary database searches primarily returned papers referencing exercise in adult lung disease. Literature pertaining to frailty in children (with or without lung disease) is sparse and there is a lack of agreement of the validity of such measurement. Clinically, transplant and rehabilitation programmes tend to segregate adult and	

	paediatric populations. The population of studies containing paediatric subjects are likely to be significantly different in their disease type and management and it may not be feasible to extrapolate their results to the adult population. For example, a significant proportion of adult lung transplant recipients will include COPD, a disease less likely to be seen in the paediatric population. Lung transplant literature and previous published systematic reviews of rehabilitation in solid organ transplant populations have focussed on adult or paediatrics as distinct populations. The author's interest is in the adult transplant population hence using inclusion criteria of over 18.	
p.9. Type of participants; given the well-known issues with organ harvesting under unethical circumstances in certain countries (using death row prisoners, political and religious prisoners, etc) and the prohibition of such activities or the publication of their results by most reputable transplantation journals, it is critical that a statement be included that no research data will be extracted from articles where it is not transparently clear that only ethical organ donation was the source of the lung transplants reported. Many guidelines now exist regarding this point, and it needs to be explicitly stated in all articles, including reviews, of transplant outcomes that such criteria were followed.	The prohibition against the use of executed prisoners' organs is explicitly directed towards China, which is one of the few countries where the use of prisoners' organs has been government-sanctioned (Rogers et al., 2019). Rogers et al (2019) state "The incremental policy therefore requires peer-reviewers and journal editors to ask consistently whether the research: (1) involved any biological material sourced from executed prisoners; (2) received Institutional Review Board (IRB) (Research Ethics Committee) approval and (3) required consent of donors." (p.2) We will therefore extract data (in addition to PROGRESS-PLUS) on country of origin. We plan to extract data regarding ethical approval gained (including IRB number and date gained) and any statement of donor consent. We will pay particular attention to studies published prior to the The Transplantation Society 2006 statement and subsequent change in policy by prominent transplant journals banning the inclusion of research reports from centres with unethical donor organs. Early scoping searches indicate most studies in this field have been completed by transplant centres in the EU or North America where rigorous organ donor ethical processes are established. However, we take this matter seriously. Where possible, we will extract and report any definitive statements regarding ethical procurement of donor organs. This information has been added to the table of data to be extracted (study design section). We will report any concerns regarding any potential unethical processes and will	Table 2

	acknowledge lack of reporting in our discussion. We have amended our protocol to address this issue and our intended additions to data extraction.	Lines 173-175
p.10. More detail is needed on the exercise variables that will be collected- equipment, specific exercises used, muscle groups trained, modality, how intensity was measured, how adherence was measured, etc.	We have added further detail into Table 2 (Data extraction plan) to increase the clarity of what exercise specific information we will extract where available/reported.	Table 2
p.11. It is not clear if you will include uncontrolled trials of multi-component interventions including exercise- for example exercise and nutrition. In such cases the independent effect of exercise cannot be determined for the outcome of frailty, which may improve due to many kinds of interventions.	Early scoping searches suggested very few multimodal interventions had been reported in the literature and within these, a high proportion of pre-post observational studies. The recent 2023 European Society of Organ Transplantation (ESOT) consensus statement on prehabilitation in solid organ transplantation reported: “To date, multimodal prehabilitation programmes that offer a combination of exercise, nutritional and psychosocial interventions, have not yet been studied in solid organ transplantation candidates. Literature is limited to studies investigating a single type of intervention” (Annema et al 2023, p.5). Multimodal interventions in this population are clinically important due the need to address the complex interaction between the physical and psychological health of the patient prior to transplantation (Annema et al 2023). We therefore feel they should be included in our review. Where these are found and reported we will include them as long there is an exercise component. We will discuss this issue and acknowledge the uncertainties of the mechanisms of specific interventions alone in our narrative synthesis.	Lines 193-202
p.11. The only surrogate markers of frailty mentioned are physical ones What about weight loss and fatigue- the other 2 components of the FFP for example? Need to specify that any validated frailty tool will be included as there are dozens in use.	The authors have chosen to look at physical frailty and we have been specific throughout our protocol that this is our focus. This is due to the complexity of frailty as a syndrome and the numerous outcome measures employed clinically/in research. We are interested in assessing the effectiveness of physical (exercise) interventions and how they are reflected in physical outcome measures. We intend to include a number of frailty outcomes including the FFP which reflects elements of weight loss and fatigue. We have therefore not specified weight loss or fatigue as individual outcomes of interest. We have adjusted the language used to reflect the use of validated outcome measures in measuring physical frailty.	From line 214

p.12. The database search is quite limited. Should also include pre-medline, Sports Discus, AMED, PEDro, Web of Science and Google Scholar for better sensitivity	We developed our search strategy, including the databases chosen, with an experienced medical information specialist and feel the breadth of databases chosen was adequate to identify any appropriate literature. Since submitting the protocol for your consideration we have begun database searches and have performed thorough checks of all reference lists (including relevant review articles) which have added no additional full text papers to include in the review. We feel confident that we can identify all relevant papers using the databases specified. We also plan to search clinical trials registries. We have also considered the amount of time available to the primary author as a part time predoctoral fellow and have therefore taken an informed but pragmatic approach to the databases selected for this review. Other BMJ Open frailty systematic reviews with 4 or fewer databases included: https://bmjopen.bmj.com/content/8/12/e024406 (3 databases) https://bmjopen.bmj.com/content/11/5/e046980 (4 databases plus unpublished pre-print databases) https://bmjopen.bmj.com/content/10/9/e037476 (3 databases)	From line 231
p.13. The search is not sensitive enough. Need to search in all fields, not just title and abstract. Other terms for exercise should be included. In the limitations you say you searched for the outcome -this does not appear to be the case in the Medine example you included? Need to state explicitly that theses and unpublished studies will not be included. Need to state that abstracts will be followed up to see if published, as well as theses.	After consultation with a medical information specialist, we adopted an approach aligned to Cochrane review search strategies and during search strategy development did not find that searching in fields beyond the title and abstract generated any further literature that met our inclusion criteria. We tested and developed our search terms iteratively and found extended terms around exercise did not increase return rates of studies for each database. We have been more explicit in exclusion of theses and unpublished studies as recommended, alongside the checking of abstracts being available in full text form. In our limitations we stated “Using outcomes as a key criterion for inclusion risks missing some relevant studies due to the potential for reporting bias.” Whilst being part of the inclusion criteria, we will not use specific outcomes as search terms as we do not want to dictate which frailty measures we will include (due to the large number of possibilities). In an attempt to reduce our susceptibility to reporting bias, we will provisionally include all studies meeting our	Lines 241-243

	inclusion criteria for population, intervention, comparator and study design. If frailty outcomes are not reported, we will contact authors to check if any were measured but excluded from the publication. We will not exclude studies based on outcomes in the first instance and will explicitly state where author contact for this reason was attempted, and whether it was successful in clarifying measurement of relevant outcomes. Thank you for highlighting that this required more clarity in the manuscript. We have rectified this.	Lines 243-247
p.14 Ethics approval needs to include the transparent reporting of the ethical nature of the organ donation process followed as this is routinely violated in publications from China.	Please see comments in response to previous reviewer comment above.	Lines 173-175 and 349 - 350
p.14 Exercise description needs to include type of equipment and specific exercises utilised, and how the intensity was measured and how the training was progressed if it was as well as any behavioural change methodology used. See CERT tool for the exercise elements that need to be included which are more specific than TIDieR for exercise studies. Adherence needs to include not only attendance, but adherence to the intensity, volume and progression principles of the prescription.	We have edited the detail of data extraction for exercise interventions to reflect the comments, specifically equipment used, exercises performed, prescription and progression methods and adherence measurement. We have paid attention to the level of detail according to the CERT tool and made reference to this where applicable.	Table 2
p.15. You have not defined the outcomes that will be included as index of frailty in the table, only how you will present them statistically.	We have added more information into the table to define outcomes to be extracted.	Table 2
p.15 by SMD, need to define the SD used- do you mean the Standardised mean response where it is the SD of the change score? Or Cohens d where it is the SD of the baseline sample?	We have consulted a medical statistician for support in identifying the appropriate approach. We have clarified the possible approaches we will adopt depending on the study designs included in the review. We are aware our approach may vary depending on the study designs included and will be clear on which approach was most appropriate and why in our final report. We will also discuss the reasons for our chosen final approach. We have edited to clarify this.	From line 267 and also 294-312
p. 17. You have not indicated how you will calculate ESs and how you will combine them if possible. What SD will you use? What will you do if only the SD of the change scores are reported, or only the post test scores are reported, and other studies report pre and post means and SDs? You will need relative ESs for the controlled trials and absolute ESs for the uncontrolled trials.	See comment above. If change from baseline data is presented without final score data, we will use an approach suggested in the Cochrane handbook (through rearrangement of a formula) to estimate the SD of the final score using an estimated correlation coefficient. Our medical statistician (added to our acknowledgements) advises against combining different study designs (ie.	From line 267 And also 294-312

	Controlled and uncontrolled) in a meta-analysis. We have attempted to plan for all situations and relevant approaches but we acknowledge that our approach may differ based on the study designs and reported results within each. We will consult again with the medical statistician to ensure we use an appropriate approach to dealing with the data throughout the data extraction and analysis process. We have edited to clarify this.	
Will you include studies that only measure frailty at post transplant assessment?	We will include outcomes measured at baseline and at any time point after the intervention (including post-transplant). We will specify whether outcomes have been assessed pre or postoperatively and will be discuss the potential confounding which could be present in the case of post-transplant measurement. Confounding by post-surgical measurement only will be addressed in our risk of bias assessment.	Lines 218-222
you have not specified when the frailty outcomes must be measured relative to the intervention. Is it only if they are immediately pre and post intervention? What about follow up post surgery? What about rehabilitation post- surgery? How will that be assessed and factored into the results? How soon after surgery can the frailty outcome be included as it is likely to change markedly in the peri-operative, acute hospital, post discharge phases. None of this is described and is critical to your findings.	We will include all time points for frailty measurement where they occur after the intervention. Preliminary searches suggest a high percentage of studies measuring outcomes immediately after the intervention has concluded, or at a fixed time point after the start of the intervention. We will include measurement of the outcomes if they occur during the post-transplant period but will be clear that this increases the likelihood of confounders and will discuss these studies separately in narrative synthesis. We will be clear where post transplant rehabilitation has occurred (in relation to the outcome measurement) and discuss the likely impact this may have on outcomes. We have clarified this in the protocol.	Lines 218-222
Reviewer: 2 Dr. Tania Janaudis-Ferreira, McGill University, Research institute of McGill University Health centre		
My main concern is about the feasibility of this review. A systematic review on exercise interventions in solid organ transplant candidates, published by Pesce de Souza et colleagues in 2020, identified only one article in lung transplant candidates (a pre-post study) that included a measure of frailty. Although the authors address the issue of limited RCTs in this field in their protocol, and mention that they will consider study designs other than RCTs, I am still concerned that the number of published articles won't be enough for their review and meta-analysis. I suggest that the authors conduct a preliminary search including all	We ran preliminary searches whilst finalising our search strategies and encountered a dearth of RCTs as a concern. We therefore extended our study design inclusion criteria to include “original primary research studies with any design, including those without controls, with more than 10 participants” (p.8). Our PROSPERO registration was amended accordingly. Where the lack of suitable studies for inclusion precludes a meta-analysis we will conduct a narrative synthesis guided by SWIM principles We felt the number of studies identified in our preliminary searches would allow for sufficient narrative synthesis, potential meta-analysis for some outcomes, and importantly, identification of the gaps in the evidence base	From line 166-171

study designs and present the results in their protocol.	to inform future study design. We expect to find between 10-20 studies when surrogate frailty measures are also accounted for.	
Another question I have is about the intervention of interest. Will the authors consider studies that included a multimodal pre-habilitation program? For example, that include a nutritional intervention alone or in addition to exercise? What about psychological interventions?	We have amended our protocol to provide more detail regarding the intended interventions to be included. “The intervention may be single (one form of exercise), multimodal (for example a combination of strength and endurance training) or multi-component (e.g. exercise and a nutritional intervention). The intervention must contain an exercise or physical activity component.” (p.9). We also plan to extract data detailing any “motivation or behavioural change strategies used” (Table 2).	Table 2 Lines 193-202
A recent consensus statement from the European Society of Organ Transplantation (ESOT) (published in July 2023) focused on multimodal pre-habilitation interventions in SOT and may be a good reference for the authors. The reference list of this consensus statement may help the authors determine the feasibility of their review. https://pubmed.ncbi.nlm.nih.gov/37547750/ [pubmed.ncbi.nlm.nih.gov] “European Society of Organ Transplantation (ESOT) Consensus Statement on Prehabilitation for Solid Organ Transplantation Candidates” by Annema and colleagues.	Our initial manuscript for submission was finalised prior to this publication. This appears to be an important paper in the field and should form part of our reasoning for the importance of this review, thank you for drawing our attention to this. We have edited the manuscript to emphasise this. This paper highlights the lack of RCTs in the solid organ transplant prehabilitation field. We therefore feel this strengthens our case for extending our study design inclusion criteria to include designs other than RCTs. This paper, alongside preliminary searches, lead us to believe the review is feasible with the wider scope of study design. This paper also highlights the importance of multi-component interventions for the solid organ transplant population. We have therefore specified we will include studies of multi-component interventions in our review (see earlier comments in table above).	Lines 114-119 Lines 193-202
What is the minimum number of studies that the authors are considering for subgroup analysis and sensitivity analysis? Based on a preliminary search, the author may be able to know of these types of analysis will be feasible	Preliminary searches and previously published literature suggest the heterogeneity of outcomes used may also limit the ability to conduct a meta-analysis however we have planned for this and will use this plan if we complete an update of this review in the future. We will consult with our medical statistician again after searches have been completed to determine if this is feasible.	n/a
Minor comment: Line 13, page 9 – a period is missing after the word “study”.	We have amended this, many thanks.	

Bibliography

Angulo J, El Assar M, Álvarez-Bustos A, et al. Physical activity and exercise: Strategies to manage frailty. *Redox Biol.* 2020;35:101513.

Annema, C., De Smet, S., Castle, E. M., Overloop, Y., Klaase, J. M., Janaudis-Ferreira, T., . . . Monbaliu, D. (2023). European Society of Organ Transplantation (ESOT) Consensus Statement on Prehabilitation for Solid Organ Transplantation Candidates [Guidelines]. *Transplant International*, 36. <https://doi.org/10.3389/ti.2023.11564>

Rogers W, Robertson MP, Ballantyne A, *et al*
Compliance with ethical standards in the reporting of donor sources and ethics review in peer-reviewed publications involving organ transplantation in China: a scoping review
BMJ Open 2019;**9**:e024473. doi: 10.1136/bmjopen-2018-024473

VERSION 2 – REVIEW

REVIEWER	Janaudis-Ferreira, Tania McGill University, School of Physical and Occupational Therapy
REVIEW RETURNED	19-Dec-2023
GENERAL COMMENTS	The authors have addressed my comments and suggestions.

VERSION 2 – AUTHOR RESPONSE